# Two-Dimensional NMR Spectroscopy of the G Protein-Coupled Receptor A_2A_AR in Lipid Nanodiscs

**DOI:** 10.3390/molecules28145419

**Published:** 2023-07-14

**Authors:** Canyong Guo, Lingyun Yang, Zhijun Liu, Dongsheng Liu, Kurt Wüthrich

**Affiliations:** 1iHuman Institute, ShanghaiTech University, Shanghai 201210, China; guocy@shanghaitech.edu.cn (C.G.); yangly@shanghaitech.edu.cn (L.Y.); liudsh@shanghaitech.edu.cn (D.L.); 2School of Life Science and Technology, ShanghaiTech University, Shanghai 201210, China; 3National Facility for Protein Science in Shanghai, ZhangJiang Lab, Shanghai Advanced Research Institute, Chinese Academy of Sciences, Shanghai 201210, China; liuzhijun@sari.ac.cn; 4Department of Integrative Structural and Computational Biology, Scripps Research, La Jolla, CA 92037, USA; 5Institute of Molecular Biology and Biophysics, ETH Zürich, Otto-Stern-Weg 5, 8093 Zürich, Switzerland

**Keywords:** GPCR, A_2A_ adenosine receptor, solution NMR, lipid nanodiscs

## Abstract

Eight hundred and twenty-six human G protein-coupled receptors (GPCRs) mediate the actions of two-thirds of the human hormones and neurotransmitters and over one-third of clinically used drugs. Studying the structure and dynamics of human GPCRs in lipid bilayer environments resembling the native cell membrane milieu is of great interest as a basis for understanding structure–function relationships and thus benefits continued drug development. Here, we incorporate the human A_2A_ adenosine receptor (A_2A_AR) into lipid nanodiscs, which represent a detergent-free environment for structural studies using nuclear magnetic resonance (NMR) in solution. The [^15^N,^1^H]-TROSY correlation spectra confirmed that the complex of [u-^15^N, ~70% ^2^H]-A_2A_AR with an inverse agonist adopts its global fold in lipid nanodiscs in solution at physiological temperature. The global assessment led to two observations of practical interest. First, A_2A_AR in nanodiscs can be stored for at least one month at 4 °C in an aqueous solvent. Second, LMNG/CHS micelles are a very close mimic of the environment of A_2A_AR in nanodiscs. The NMR signal of five individually assigned tryptophan indole ^15^N–^1^H moieties located in different regions of the receptor structure further enabled a detailed assessment of the impact of nanodiscs and LMNG/CHS micelles on the local structure and dynamics of A_2A_AR. As expected, the largest effects were observed near the lipid–water interface along the intra- and extracellular surfaces, indicating possible roles of tryptophan side chains in stabilizing GPCRs in lipid bilayer membranes.

## 1. Introduction

Eight hundred and twenty-six human G protein-coupled receptors (GPCRs) constitute the largest superfamily of integral membrane proteins (IMPs) in the human genome, which share a common architecture of seven transmembrane α-helices linked by extracellular and intracellular loops [1,2,3]. GPCRs are involved in a wide range of physiological processes by transducing signals across cellular membranes; they are therefore central to drug discovery programs, being targeted by about 35% of the currently marketed prescription pharmaceuticals [4]. In this context, GPCRs have been subject to intense efforts from structural biology [5,6,7]. X-ray crystallography and cryo-electron microscopy (cryo-EM) yielded more than 150 unique GPCR structures with over 1000 different ligands bound, as annotated in the GPCRdb [8]. The vast majority of these structural studies obtained data for GPCRs reconstituted in detergent micelles [9]. To assess the functional significance of the data thus obtained, additional studies pursued with phospholipid bilayer nanodiscs [10,11], for example, in structure determinations of the D2 dopamine receptor (DRD2)–G protein complex [12], the M2 muscarinic receptor (M2R)–β-arrestin complex [13], and the neurotensin receptor 1 (NTSR1)–G protein complex [14]. Accumulating evidence has shown that phospholipids participate in the signaling activities of GPCRs, as well as in the selectivity of G-protein coupling [15,16,17,18]. This study bears on the use of nuclear magnetic resonance (NMR) for studies of GPCRs embedded in planar lipid bilayer membranes in solution, which has been previously applied, for example, with the β_2_-adrenergic receptor (β_2_AR) [19,20,21], the A_2A_ adenosine receptor (A_2A_AR) [22,23,24,25,26], the neurotensin receptor 1 (NTSR1) [27,28], the α_1B_-adrenergic receptor (α_1B_-AR) [29], the leukotriene B_4_ receptor 2 (BLT2) [30], and the neurokinin 1 receptor (NK1R) [31]. These studies indicated that the conformational landscapes and the exchange rates between simultaneously populated conformational substates of a GPCR may be distinctly different in detergent micelles and lipid bilayers [19,20,23]. Specifically, there are indications that the charge properties of lipid head groups may modulate the structural plasticity of a GPCR in concert with orthosteric ligands [24].

In continuation of previous work on the many-parameter characterization of signaling-related structural dynamics of A_2A_AR in lauryl maltose neopentyl glycol (LMNG)/cholesteryl hemisuccinate (CHS) micelles using 2D [^15^N,^1^H]-TROSY (transverse relaxation-optimized spectroscopy [32]) NMR in solution [33,34], we reconstituted doubly labeled A_2A_AR into phospholipid nanodiscs for NMR studies. Well-resolved 2D [^15^N,^1^H]-TROSY correlation spectra of the complex of [u-^15^N, ~70% ^2^H]-A_2A_AR with the inverse agonist ZM241385 in MSP1D1ΔH5 (1-palmitoyl-2-oleoyl-*sn*-glycero-3-phosphocholine (POPC)/1-palmitoyl-2-oleoyl-*sn*-glycero-3-phospho-L-serine (POPS) 7:3) nanodiscs opened the door for initial comparisons of the receptor complex in detergent micelles and lipid nanodiscs. GPCR preparations yielding high-quality multidimensional NMR spectra in nanodisc milieus provide an attractive avenue to study the signaling-related conformational dynamics of GPCRs and their interactions with downstream partner proteins in a detergent-free membrane environment.

## 2. Results and Discussion

### 2.1. Preparation and Characterization of A_2A_AR in Nanodiscs

*Pichia pastoris* was selected as the host for heterologous expression of human A_2A_AR, as it allows cost-effective production of uniformly ^2^H, ^13^C, ^15^N-labeled eukaryotic membrane proteins [35,36,37,38]. The wild-type-like A_2A_AR construct used in this study (Figure 1a) is identical to the one previously used by Eddy et al. [33]. A_2A_AR was extracted from the cell membrane, solubilized in LMNG/CHS (20:1) micelles, and purified using immobilized metal affinity chromatography (IMAC) with Co^2+^ (Figure 1b, lane 1). The purified A_2A_AR in detergent micelles was then reconstituted into lipid nanodiscs formed with the membrane scaffold protein MSP1D1ΔH5 and containing a 7:3 lipid ratio of the zwitterionic phospholipid POPC and the anionic lipid POPS; this mixture of zwitterionic and anionic lipids is commonly used in the reconstitution of GPCRs in nanodiscs [24,26,31]. The small membrane scaffold protein MSP1D1ΔH5 was chosen for this study because the associated reduced size of the A_2A_AR-containing lipid nanoparticles could be expected to provide advantages for multidimensional NMR studies [39,40]. The resulting A_2A_AR-containing nanodiscs were separated from the empty nanodiscs with IMAC with Ni^2+^ (Figure 1b, lanes 2 and 3), and the size exclusion chromatography (SEC) revealed that the resulting purified A_2A_AR in nanodiscs was monodisperse (Figure 1c).

### 2.2. Complex of A_2A_AR with an Inverse Agonist Adopts Similar Global Folds in Phospholipid Nanodiscs and LMNG/CHS Micelles

A 2D [^15^N,^1^H]-TROSY correlation spectrum of the complex of [u-^15^N, ~70% ^2^H]-A_2A_AR with the inverse agonist ZM241385 in LMNG/CHS (20:1) micelles was recorded at 37 °C (Figure 2a). In the data analysis, we closely followed the procedure used by Eddy et al. [33]. Briefly, 30 cross peaks in the backbone ^15^N–^1^H groups, which are subject to large conformation-dependent chemical shifts [42] and are therefore well-resolved in the 2D NMR spectra, were used to assess the global folds of the GPCR in different milieus. This overall spectral comparison is supplemented in Section 2.3 with information on localized conformational changes that are accessible using observation of individually assigned signals of tryptophan indole ^15^N–^1^H groups. Comparison of the overall spectra, with a focus on the resonances of previously assigned tryptophan indole ^15^N–^1^H moieties [33] and other well-dispersed ^15^N–^1^H cross peaks, showed near-identity of the spectrum in Figure 2a and previously published corresponding data [33], showing that we successfully reproduced the previously described 2D [^15^N,^1^H]-TROSY correlation spectrum of A_2A_AR in LMNG/CHS micelles [33]. The A_2A_AR in detergent micelles used to record the spectrum shown in Figure 2a were then transferred into lipid nanodiscs. The complex of [u-^15^N, ~70% ^2^H]-A_2A_AR with the inverse agonist ZM241385 in MSP1D1ΔH5 (POPC/POPS 7:3) nanodiscs yielded the 2D [^15^N,^1^H]-TROSY correlation spectrum shown in Figure 2b. The aforementioned selection of 30 well-resolved ^15^N–^1^H cross-peaks provided the basis for an assessment of the overall global folds of the receptor in the detergent micelles and in lipid nanodiscs, respectively (Figure 2, Table 1). Among these 30 not individually assigned backbone ^15^N–^1^H signals in the spectrum shown in Figure 2a, all but 4 could be matched with a peak at a ^1^H-chemical shift difference smaller than 0.10 ppm in the spectrum shown in Figure 2b, with 18 of the corresponding peak pairs showing ^1^H-chemical shift differences smaller than 0.05 ppm (Table 1). The four outliers were peaks 2, 4, 14, and 28, with chemical shift differences of 0.20, 0.14, 0.12, and 0.12 ppm, respectively. Since the group of 30 signals was selected for their large conformation-dependent shifts, a major structural rearrangement of the A_2A_AR fold when going from reconstitution in LMNG/CHS micelles to nanodiscs is highly unlikely [42]. The average linewidths at half-height of the 30 ^15^N–^1^H cross peaks (Table 1) in POPC/POPS nanodiscs (49 ± 11 Hz in the ^1^H dimension; 41 ± 6 Hz in the ^15^N dimension) were also similar to those in LMNG/CHS micelles (47 ± 9 Hz in the ^1^H dimension; 41 ± 5 Hz in the ^15^N dimension). These data imply that the receptor embedded either in the nanodiscs or LMNG/CHS micelles has the same global fold and is under closely similar motional regimes. Remarkably, the nanodisc sample remained stable for at least one month at 4 °C, as confirmed with repetition of the 2D [^15^N,^1^H]-TROSY correlation experiment (Figure 3a). Figure 3a shows identical patterns in peaks 1 to 30 as in Figure 2b, documenting that the chemical shifts in these peaks were faithfully conserved for one month. Figure 3b further shows that the intensity distribution among these 30 peaks was also preserved. The long-time stability of the A_2A_AR preparation in nanodiscs is thus documented with the multi-parameter details that the 2D NMR spectra afford.

### 2.3. Discrete Localized Impact of Different Membrane Mimetics from the Observation of Individually Assigned Tryptophan Indole ^15^N–^1^H NMR Signals in A_2A_AR

Previously, the resonances of the tryptophan indole ^15^N–^1^H moieties of A_2A_AR in LMNG/CHS micelles were assigned with single-residue amino acid replacements [33]. Here, corresponding resonances in POPC/POPS nanodiscs were tentatively assigned using chemical shift matching with the resonances of A_2A_AR in LMNG/CHS micelles. Serving as discrete NMR probes, these signals enabled the addition of local details to the global structure comparison performed in the preceding section.

It became readily apparent that the individual indole ^15^N–^1^H signals were influenced by the membrane mimetics in widely different, molecular region-dependent manners. The chemical shifts W246^6.48^ (the superscript refers to the Ballesteros–Weinstein nomenclature [44]) in LMNG/CHS micelles and nanodiscs are very similar (Figure 4b), although they have a very large conformation-dependent ^1^H chemical shift contribution [42] that placed them outside of the characteristic low-field tryptophan indole ^15^N–^1^H spectral region (Figure 4a) [42]. W^6.48^ is described in the GPCR literature as the “toggle switch tryptophan” because it undergoes major structural rearrangements during GPCR activation [45]. W246^6.48^ is located at the bottom of the orthosteric ligand binding groove (Figure 5a), and because of the unusual ^1^H chemical shift (Figure 4a), its indole ^15^N–^1^H signal is highly sensitive to local conformational rearrangements [42], such as those triggered by bound ligands of variable efficacies. The very small downfield ^1^H shift of 0.04 ppm when changing the membrane mimetic from LMNG/CHS micelles to lipid nanodiscs indicates that W246^6.48^ is amazingly well shielded from the impact of environmental variations.

W29^1.55^ was located near the intracellular end of the trans-membrane helix 1 (TM1) (Figure 1a and Figure 5a) and had almost identical chemical shifts in the indole ^15^N–^1^H signal for A_2A_AR in LMNG/CHS micelles or in nanodiscs, but the signal in the micelles was broadened to 109 Hz as compared to the linewidth of 38 Hz in nanodiscs (Figure 4b). The line broadening in the detergent micelles implies the co-existence of multiple side chain conformations, which coincides with observations in the W29^1.55^ signal reported in a previous study [33]. In contrast, a single conformational species of W29^1.55^ was stabilized in nanodiscs; W29^1.55^ thus represents a situation where the membrane environment strongly modulates the local conformational plasticity of the receptor.

In the crystal structure of A_2A_AR in complex with ZM241385 [46], W32^1.58^, W143^TM4/ECL2,^ and W268^7.33^ were all located near the membrane–water interface, with the side chains oriented towards the hydrophobic membrane mimetic. It was noticed elsewhere that tryptophans tend to be located at the bilayer interface of α-helical membrane proteins, where the ordered lipids provide anchor points for interfacial tryptophans through multiple types of molecular interactions [47,48,49]. Thereby, the indole ^15^N–^1^H moiety can act as a hydrogen bond donor and form hydrogen bonds with phospholipid headgroups, making indole ^15^N–^1^H NMR probes at the lipid bilayer–water interface sensitive to changes in the membrane mimetics. The ^1^H chemical shift differences between LMNG/CHS micelles and POPC/POPS nanodiscs of 0.25, 0.07, and 0.20 ppm, respectively, for W32^1.58^, W143^TM4/ECL2^, and W268^7.33^ (Figure 4b) can be rationalized using the implicated interactions with lipids in these interfacial tryptophan residues.

## 3. Materials and Methods

### 3.1. A_2A_AR Expression and Purification

The construct used in this study contained the sequence of residues 1–316, with the potential N-glycosylation site at Asn154 mutated to Gln, an N-terminal Flag tag, and a C-terminal 10x His tag (Figure 1a). The receptor was expressed in *Pichia pastoris* strain BG12 (bioGcrammatics, Inc., Carlsbad, CA, USA). To produce a uniformly ^15^N-labeled protein, we grew cultures in a D_2_O-based medium containing 3 g/L of ^15^NH_4_Cl (Cambridge Isotope Laboratories) as the sole nitrogen source. To adapt the yeast cells for growth in D_2_O-containing media, they were first grown in 4 mL buffered minimal glycerol media (BMGY) (4.25 g/L YNB without amino acids and ammonium sulfate, 11.7 g/L NaH_2_PO_4_, 7.5 g/L Na_2_HPO_4_, 3 g/L^15^NH_4_Cl, 20 g/L glycerol, and 2.5 mL 0.02% (*w*/*v*) biotin) containing ~80% D_2_O at 30 °C and 165 rpm for two days. The cultures were centrifuged at 1174× g for 7 min, and the cells were resuspended in 50 mL BMGY medium (4.25 g/L YNB without amino acids and ammonium sulfate, 11.7 g/L NaH_2_PO_4_, 7.5 g/L Na_2_HPO_4_, 3 g/L^15^NH_4_Cl, 20 g/L glycerol, and 2.5 mL 0.02% (*w*/*v*) biotin) containing 99.9% D_2_O and shaken at 30 °C and 165 rpm for another two days. The cultures were centrifuged at 1174× g for 10 min and resuspended in 500 mL BMGY medium (4.25 g/L YNB without amino acids and ammonium sulfate, 11.7 g/L NaH_2_PO_4_, 7.5 g/L Na_2_HPO_4_, 3 g/L^15^NH_4_Cl, 22 g/L glycerol, and 2.5 mL 0.02% biotin) containing 99.9% D_2_O. The large-scale cultures were shaken at 30 °C and 165 rpm until the glycerol was completely consumed. The cells were centrifuged at 1529× g for 10 min and resuspended in methanol-deprived buffered minimal glycerol media (BMMY) (4.25 g/L YNB without amino acids and ammonium sulfate, 11.7 g/L NaH_2_PO_4_, 7.5 g/L Na_2_HPO_4_, 3 g/L^15^NH_4_Cl, and 2.5 mL 0.02% (*w*/*v*) biotin) containing 99.9% D_2_O. After 6 h of starvation and shaking at 27 °C and 165 rpm, protein expression was induced with the addition of 5 g/L of methanol per 13 h. In addition, 1 mM theophylline was added to the BMMY cultures during the first methanol induction; theophylline acts as a weak antagonist for stabilizing A_2A_AR in the cell membrane. The cells were harvested after 40 h of induction at 27 °C using centrifugation at 3470× g for 10 min and stored at −80 °C.

For A_2A_AR purification, the cell pellets were resuspended in lysis buffer (50 mM sodium phosphate pH 7.0, 100 mM NaCl, 5% (*w*/*v*) glycerol, EDTA-free protease inhibitor cocktail (MedChemExpress)) and disrupted using a high-pressure cell disruptor for 15 min at 1100 bar and a temperature of 4 °C. The cell membranes were isolated using centrifugation at 142,400× *g* for 40 min and then resuspended in a high-salt buffer (10 mM HEPES pH 7.0, 10 mM MgCl_2_, 20 mM KCl, 1 M NaCl, EDTA-free protease inhibitor cocktail) containing 4 mM theophylline. The resuspended membranes were incubated with 2 mg/mL iodoacetamide (Sigma, St. Louis, MO, USA) for 1 h at 4 °C and then mixed with the same volume of solubilization buffer (90 mM HEPES pH 7.0, 1% (*w*/*v*) LMNG (Anatrace), 0.05% CHS (Sigma, St. Louis, MO, USA)). After incubation at 4 °C for 6 h, the supernatant was isolated using centrifugation at 142,400× *g* for 40 min and incubated overnight at 4 °C with TALON metal affinity resin (Clontech) and 30 mM imidazole. The resin was first washed with 20 column volumes of wash buffer 1 (25 mM HEPES pH 7.0, 500 mM NaCl, 10 mM MgCl_2_, 0.1% LMNG, 0.0025% CHS, 8 mM ATP, 30 mM imidazole), then with 40 column volumes of wash buffer 2 (25 mM HEPES pH 7.0, 250 mM NaCl, 0.05% LMNG, 0.0025% CHS, 5% glycerol, 30 mM imidazole, 20 µM ZM241385), and eluted with 10 column volumes of elution buffer (25 mM HEPES pH 7.0, 250 mM NaCl, 0.05% LMNG, 0.0025% CHS, 5% glycerol, 300 mM imidazole, 25 µM ZM241385). The eluted A_2A_AR was concentrated to 500 µL using a 50 kDa molecular weight cut-off centrifugal filter (Amicon Ultra-15, Merk Millipore Co., Ltd., Cork, Ireland) at 227× *g* and exchanged into NMR buffer (20 mM sodium phosphate pH 6.8, 100 mM NaCl, 0.1 mM EDTA, 50 µM ZM241385) with three rounds of dilution and concentration.

### 3.2. MSP1D1ΔH5 Expression and Purification

The pET28a plasmid encoding for MSP1D1ΔH5, with a N-terminal 6x His tag followed by a tobacco etch virus (TEV) protease recognition site, was purchased from Addgene (plasmid #71714). The protein was expressed in *E. coli* strain BL21 (DE3) cells. The cells were grown in Terrific Broth (TB) media supplemented with 50 µg/mL kanamycin. The cell cultures were grown at 37 °C and 220 rpm to an optical density at 600 nm of 0.6–0.8 (typically attained after about 3 to 4 h). Protein expression was then induced with the addition of 1 mM IPTG at 37 °C for 4 h. The cells were harvested using centrifugation at 7808× *g* for 15 min. 

For MSP1D1ΔH5 purification, the cell pellets were resuspended in buffer A (50 mM Tris-HCl pH 7.76 at 5 °C, 300 mM NaCl) supplemented with 0.02% (*v*/*v*) Triton X-100 and EDTA-free protease inhibitor cocktail and disrupted using a high-pressure cell disruptor operating at 1000 bar for 10 min. After centrifugation at 26,916× *g* for 10 min at 4 °C, the supernatant was supplemented with 10 mM MgCl_2_ and a small amount of deoxyribonuclease I (Sigma, St. Louis, MO, USA) and stirred at room temperature for 10 min. The supernatant was then applied to a Ni-NTA resin at 4 °C. The resin was washed with 10 column volumes of wash buffer 1 (buffer A + 1% Triton X-100), 10 column volumes of wash buffer 2 (buffer A + 50 mM sodium cholate), 10 column volumes of wash buffer 3 (buffer A), 10 column volumes of wash buffer 4 (buffer A + 10 mM imidazole) and eluted with 5 column volumes of elution buffer (buffer A + 300 mM imidazole). The His tag was cleaved with TEV protease, and the protein was dialyzed overnight at 4 °C against 2 L of buffer A, using 3.5 kDa MWCO dialysis tubes (Thermo, Rockford, IL, USA). The protein was applied to a Ni-NTA resin (equilibrated with buffer A), and the resin was washed with 10 column volumes of buffer A. The MSP1D1ΔH5 in the flow-through was concentrated using a 10 kDa molecular weight cut-off centrifugal filter and stored at −80 °C.

### 3.3. Nanodiscs Assembly and Purification

The purified A_2A_AR in LMNG/CHS micelles and the MSP1D1ΔH5 were quantified using bicinchoninic acid (BCA) assays. A_2A_AR, MSP1D1ΔH5, and cholate-solubilized lipids were mixed at a molar ratio of 1:5:300, with a final concentration of 20 mM sodium cholate. The lipid mixture contained 1-palmitoyl-2-oleoyl-*sn*-glycero-3-phosphocholine (POPC) (Avanti) and 1-palmitoyl-2-oleoyl-*sn*-glycero-3-phospho-L-serine sodium salt (POPS) (Avanti) with a molar ratio of 7:3. The mixture was incubated at 4 °C for 1 h with gentle agitation, followed by the addition of 50% (*v*/*v*) Bio-Beads SM-2 (Bio-Rad, Hercules, CA, USA), and further incubated at 4 °C overnight with gentle agitation. The empty nanodiscs were removed by passing through Ni-NTA resin, and the resin was washed with 10 column volumes of wash buffer (25 mM HEPES pH 7.0, 150 mM NaCl, 20 µM ZM241385). The A_2A_AR in nanodiscs was eluted with 10 column volumes of elution buffer (20 mM HEPES pH 7.0, 75 mM NaCl, 20 µM ZM241385) and concentrated to 500 µL using a 50 kDa molecular weight cut-off centrifugal filter at 227× *g*. The purified A_2A_AR in nanodiscs were exchanged to NMR buffer (20 mM sodium phosphate pH 6.8, 100 mM NaCl, 0.1 mM EDTA, 50 µM ZM241385) with three rounds of dilution and concentration. 

### 3.4. NMR Experiments

NMR experiments were performed using a Bruker Avance 900 MHz spectrometer equipped with a 5 mm TCI cryogenically cooled probe at 37 °C. For NMR experiments, samples of the complex of [u-^15^N, ~70% ^2^H]-A_2A_AR with the inverse agonist ZM241385 either in LMNG/CHS (20:1) micelles or in MSP1D1ΔH5 (POPC/POPS 7:3) nanodiscs were prepared in NMR buffer (20 mM sodium phosphate pH 6.8, 100 mM NaCl, 0.1 mM EDTA, 50 µM ZM241385) containing 5% D_2_O. Then, 2D [^15^N,^1^H]-TROSY correlation spectra were recorded with 2048 points in the ^1^H dimension and 160 points in the ^15^N dimension. The 2D TROSY spectrum of A_2A_AR in micelles was collected with 576 scans and of A_2A_AR in nanodiscs with 624 scans. All NMR data were processed with NMRPipe [50] and analyzed with Poky [43].

## 4. Conclusions

High-quality 2D [^15^N,^1^H]-TROSY correlation NMR spectra of the complex of [u-^15^N, ~70% ^2^H]-A_2A_AR with the inverse agonist ZM241385 in LMNG/CHS (20:1) micelles and MSP1D1ΔH5 (POPC/POPS 7:3) nanodiscs were obtained at physiological temperature. This A_2A_AR complex maintains its global fold in the phospholipid bilayer nanodiscs for at least 30 days at 4 °C, which opens the door for extensive structural investigations using multidimensional NMR. Five individually assigned tryptophan indole ^15^N–^1^H moieties distributed throughout the receptor provided observations of the localized impact of the detergent micelles and lipid nanodiscs. Specifically, indole ^15^N–^1^H groups near the lipid–water interfaces displayed larger chemical shift changes than those in the A_2A_AR core, suggesting a structure-stabilizing role of the interfacial tryptophan residues (Figure 5b).

The vast majority of NMR studies on the structure and function of GPCRs have so far used extrinsic probe molecules that are typically attached to amino acid side chains near the surface of the receptors [51]. The present work presents a way to add many-parameter NMR characterization of GPCR activation in a detergent-free environment to these single-parameter approaches. The use of A_2A_AR as a vehicle for this technical development provides interesting perspectives for future work in cancer research. A_2A_AR was shown to be the major target for influencing the immunosuppressive effects of adenosine in tumor microenvironments [52,53]. Detailed structural characterization of A_2A_AR is an ongoing part of projects aimed at developing new cancer drugs that target this GPCR [54,55].

## Figures and Tables

**Figure 1 molecules-28-05419-f001:**
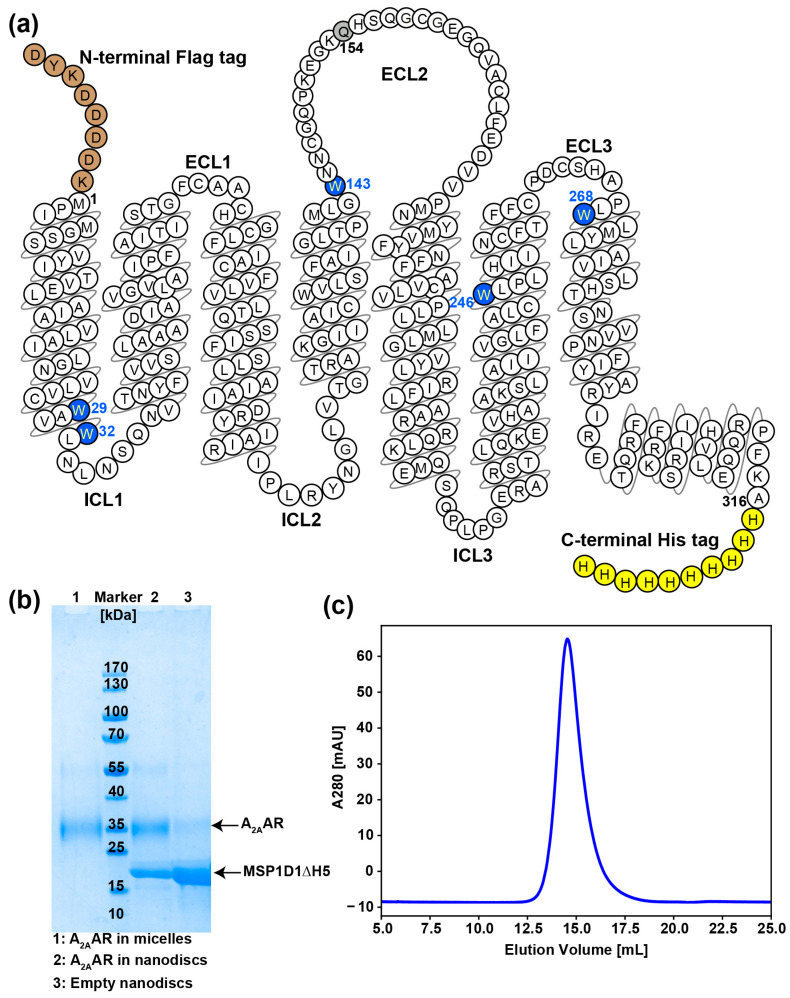
Chemical structure and biochemical characterization of A_2A_AR in lipid nanodiscs. (**a**) Snake plot modified from the GPCRdb (https://gpcrdb.org/, 1 June 2023) [41] showing the construct of the truncated A_2A_AR (1–316) with an N-terminal Flag tag (brown), one mutation, N154Q (grey), and a C-terminal 10× His tag (yellow) used. Five tryptophan residues that are used as NMR probes in this study are colored blue, with the sequence numbers indicated in blue. (**b**) SDS-PAGE gel analysis of the purified A_2A_AR in MSP1D1ΔH5 (POPC/POPS 7:3) nanodiscs (lane 2) using immobilized metal affinity chromatography (IMAC). The theoretical molecular weights of A_2A_AR and MSP1D1ΔH5 are 37.6 and 19.5 kDa, respectively. (**c**) Size exclusion chromatogram of purified A_2A_AR in MSP1D1ΔH5 (POPC/POPS 7:3) nanodiscs after IMAC.

**Figure 2 molecules-28-05419-f002:**
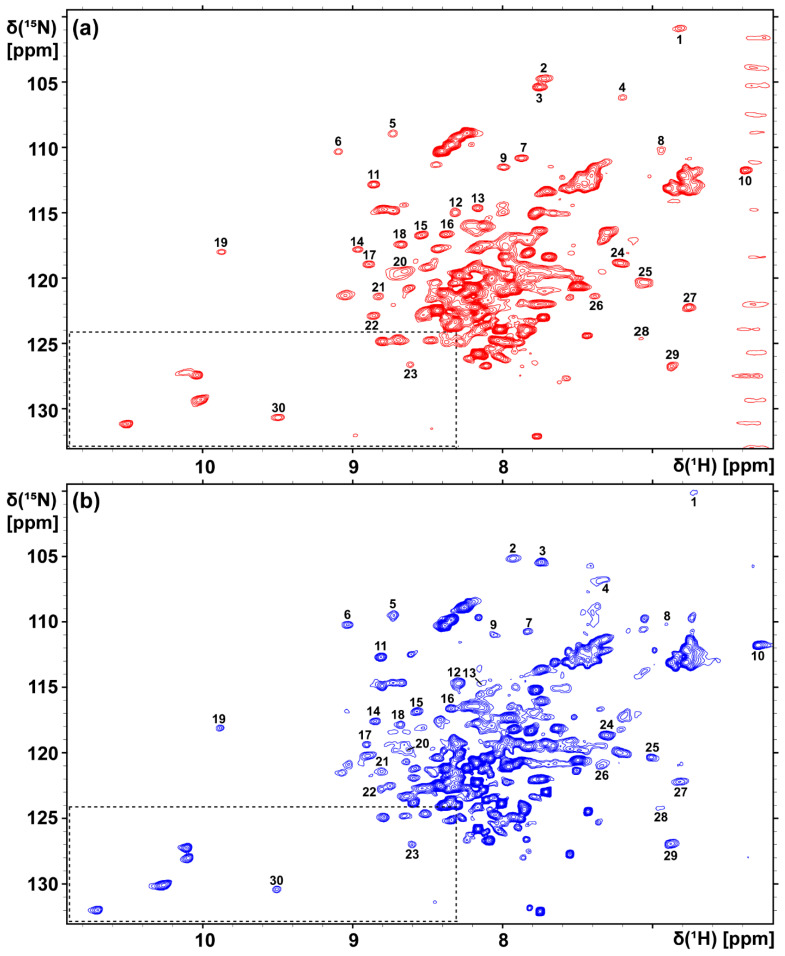
Two-dimensional [^15^N, ^1^H]-TROSY correlation spectra of the complex of [u-^15^N, ~70% ^2^H]-A_2A_AR with the inverse agonist ZM241385 in LMNG/CHS (20:1) micelles (**a**) and in MSP1D1ΔH5 (POPC/POPS 7:3) nanodiscs (**b**). The spectra were recorded at pH 6.8 and 37 °C. In (**a**), 30 previously discussed ([33], see text), well-resolved ^15^N–^1^H cross-peaks are identified with numbers; in (**b**), 30 corresponding peaks are identified based on their near-identical chemical shifts. These polypeptide backbone ^15^N–^1^H signals were used to monitor possible variations in the global folds of A_2A_AR in the two different environments. The spectral region outlined with broken lines in the spectra contains the area that usually includes the signals of the Trp indole ^15^N–^1^H groups: it was extended along the ^1^H chemical shift axis to a higher field so as to include also the signal of Trp 246 at its unusual ^1^H chemical shift (see also Figure 4a).

**Figure 3 molecules-28-05419-f003:**
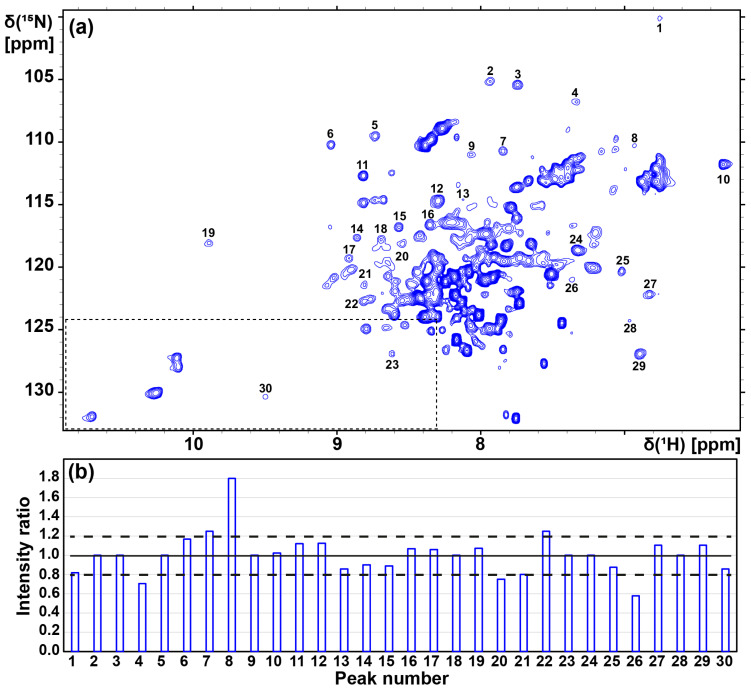
Long-term stability of the complex of [u-^15^N, ~70% ^2^H]-A_2A_AR with the inverse agonist ZM241385 in MSP1D1ΔH5 (POPC/POPS 7:3) nanodiscs. (**a**) The 2D [^15^N, ^1^H]-TROSY correlation spectrum was recorded with the same sample that was used in Figure 2b after it was stored at 4 °C for 1 month. The spectral region outlined with broken lines in the spectrum shown in (**a**) contains the area that usually includes the signals of the Trp indole ^15^N–^1^H groups: it was extended along the ^1^H chemical shift axis to a higher field so as to include also the signal of Trp 246 at its unusual ^1^H chemical shift (see also Figure 4a). (**b**) Intensity ratios of the peaks numbered 1 to 30 in Figure 2b and Figure 3a. Intensity ratios were calculated by dividing the peak intensities in Figure 3a by the corresponding peak intensities in Figure 2b. Each peak intensity was calibrated against the noise level using the software Poky [43]. The black solid (dashed) lines indicate the average (±1 standard deviation) intensity ratio.

**Figure 4 molecules-28-05419-f004:**
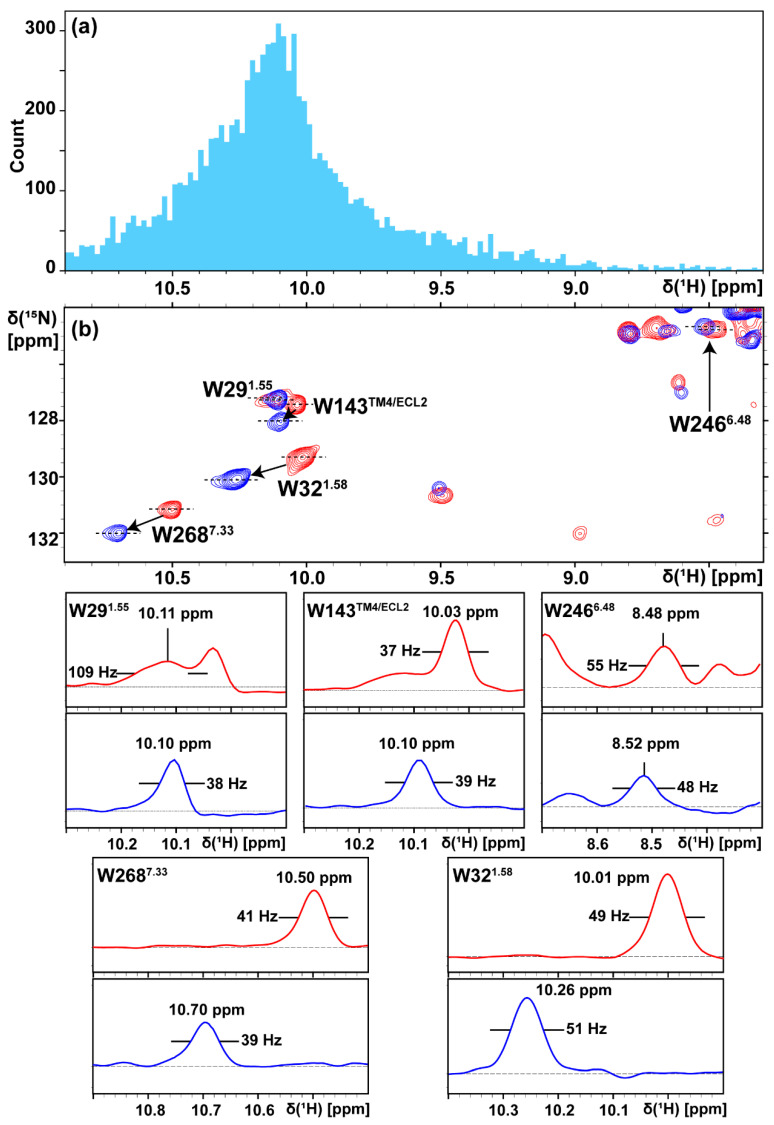
(**a**) Indole ^15^N–^1^H proton chemical shift distribution of ~9100 tryptophan residues in the Biological Magnetic Resonance Bank (BMRB) (https://bmrb.io/, accessed on 23 April 2023). (**b**) Overlay of the spectral region containing tryptophan indole ^15^N–^1^H signals (dotted rectangle in Figure 2a,b and Figure 3a) of the complex of [u-^15^N, ~70% ^2^H]-A_2A_AR with the inverse agonist ZM241385 in LMNG/CHS (20:1) micelles (red) and MSP1D1ΔH5 (POPC/POPS 7:3) nanodiscs (blue). For each residue, cross-sections along the ^1^H chemical shift axis at the dashed lines are shown in the bottom five panels.

**Figure 5 molecules-28-05419-f005:**
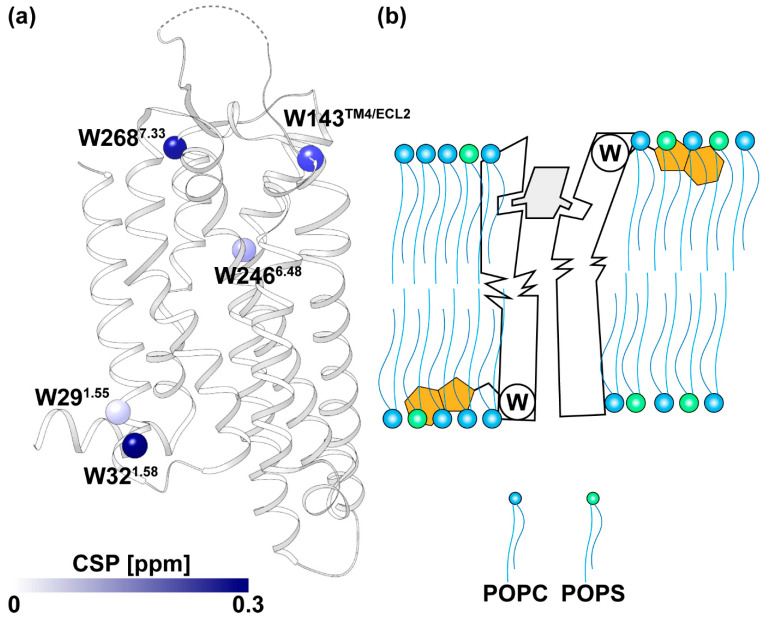
(**a**) Side view showing the A_2A_AR complex with ZM241385 (PDB code 3PWH [45]). Tryptophan residues are represented as spheres and colored according to their chemical shift changes between LMNG/CHS (20:1) micelles and MSP1D1ΔH5 (POPC/POPS 7:3) nanodiscs, with larger differences represented with darker coloring. (**b**) Schematic drawing showing how tryptophan residues at the water–lipid interface might stabilize A_2A_AR in the lipid bilayer membrane. The tryptophan indole rings are brown, and a grey shape indicates a bound orthosteric ligand.

**Table 1 molecules-28-05419-t001:** The ^1^H–^15^N chemical shifts and linewidths of 30 well-resolved NMR signals of the complex of [u-^15^N, ~70% ^2^H]-A_2A_AR with the inverse agonist ZM241385 in LMNG/CHS (20:1) micelles and in MSP1D1ΔH5 (POPC/POPS 7:3) nanodiscs (Figure 2 and Figure 3).

Peak Number ^1^	LMNG/CHS Micelles (Figure 2a)				POPC/POPS Nanodiscs (Figure 2b)			
	^1^H		^15^N		^1^H		^15^N	
	δ [ppm]	ν_1/2_ (Hz)	δ [ppm]	ν_1/2_ (Hz)	δ [ppm]	ν_1/2_ (Hz)	δ [ppm]	ν_1/2_ (Hz)
1	6.82	66	100.88	40	6.73	53	100.15	43
2	7.73	65	104.75	40	7.93	67	105.15	43
3	7.76	44	105.38	39	7.74	49	105.43	40
4	7.20	51	106.19	46	7.34	94	106.82	45
5	8.73	43	108.95	51	8.73	43	109.52	53
6	9.10	40	110.33	41	9.03	47	110.24	39
7	7.88	51	110.82	36	7.84	43	110.74	35
8	6.94	39	110.24	50	6.91	38	110.19	34
9	8.00	54	111.53	38	8.06	42	110.99	38
10	6.39	41	111.77	37	6.30	41	111.80	37
11	8.86	39	112.84	40	8.81	42	112.71	39
12	8.32	42	115.00	48	8.29	51	114.71	49
13	8.17	40	114.65	41	8.15	44	114.74	43
14	8.97	42	117.81	38	8.85	48	117.61	37
15	8.54	50	116.72	38	8.57	46	116.85	38
16	8.38	45	116.66	38	8.34	45	116.63	38
17	8.89	44	118.93	40	8.91	39	119.39	38
18	8.68	43	117.42	39	8.68	44	117.85	40
19	9.87	40	118.01	37	9.88	39	118.11	37
20	8.65	69	119.47	51	8.63	58	119.78	65
21	8.83	44	121.38	47	8.81	57	121.43	48
22	8.86	48	122.88	42	8.81	45	122.77	42
23	8.62	36	126.63	45	8.60	43	127.01	42
24	7.22	62	118.85	40	7.31	57	118.66	39
25	7.05	48	120.34	41	7.01	43	120.33	34
26	7.39	40	121.39	32	7.34	58	121.03	39
27	6.76	46	122.24	38	6.82	62	122.22	38
28	7.08	54	124.66	45	6.96	44	124.23	38
29	6.88	40	126.76	45	6.88	49	126.96	42
30	9.50	52	130.67	39	9.51	44	130.40	40

^1^ The cross peaks are identified in Figure 2.

## Data Availability

The data presented in this study are available upon request from the corresponding author.

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
