# Peer review of "Two-Dimensional NMR Spectroscopy of the G Protein-Coupled Receptor A2AAR in Lipid Nanodiscs"

_molecules, 2023, doi:10.3390/molecules28145419_

Round 1

Reviewer 1 Report

The article presents the spectrum of the human A2A adenosine receptor in lipid nanodiscs and compares it with the spectrum in mixed LMNG/CHS micelles.

In general, the paper is clear and well structured, however it lacks scientific interest. It only shows that it is possible to acquire these experiments under these conditions; which could, perhaps, be considered a technical achievement, but does not provide any new insight into the system.

The greatest interest is, as the authors mention, the opportunity to use these nanodiscs in future studies.

Figure 2, which is the main data, should be presented as an overlay of the two spectra. In this way it would be possible to compare all the signals and not just the 30 “well” dispersed peaks.

The similarities and differences would be clearer.

Figure 3 is of minimal interest, it only shows that the sample still has a signal after being stored at 4C.

What is the variation of signal intensity between the two spectra? This would help us understand how much protein is being lost.

Panel a of Figure 4 should be omitted. It is not necessary to show the histogram.

In 4b, in the traces, there are problems in the baseline (or phase correction), which in turn would cause problems when calculating the intensity at half height.

Author Response

We thank the three reviewers for the careful reading of our manuscript. We appreciate the criticisms and comments that have been advanced, and we hope that our detailed response in this letter can convince the editor of the importance of our work and result in acceptance of the paper for publication as an article in Molecules.

Reviewer 1

The article presents the spectrum of the human A2A adenosine receptor in lipid nanodiscs and compares it with the spectrum in mixed LMNG/CHS micelles.

In general, the paper is clear and well structured, however it lacks scientific interest. It only shows that it is possible to acquire these experiments under these conditions; which could, perhaps, be considered a technical achievement, but does not provide any new insight into the system.

The greatest interest is, as the authors mention, the opportunity to use these nanodiscs in future studies.

Response:

We fully understand that the reviewer identifies limitations in the scope of our publication, and all the more we are glad to read that he sees the interest of our current results with regard to continued research on structure–function studies GCPRs and possible other membrane proteins. In this regard, we have probably not given sufficient emphasis to the importance of the observation (Figure 2, Table 1) that LMNG/CHS micelles provide an environment for A2AAR that is surprisingly close to the environment in nanodiscs. This has not been shown previously on the level of detail that is afforded by multi-dimensional NMR experiments. In further response to the reviewers’ comment, we have added a sentence to the abstract of the paper (Lines 22–25 in the manuscript).

Figure 2, which is the main data, should be presented as an overlay of the two spectra. In this way it would be possible to compare all the signals and not just the 30 “well” dispersed peaks.

The similarities and differences would be clearer.

Response:

In overlays of the two spectra, there is extensive signal interference and we therefore decided to present the two spectra side-by-side, supplemented with Table 1. 30 well-dispersed backbone 15N–1H signals are used to monitor possible differences between the global folds of A2AAR in different membrane mimetics. Because of their large conformation-dependent 1H chemical shifts, these signals are largely sensitive to structure variations (see also Eddy et al., Cell 2018, 172, 68–80). The table 1 documents the near-identical chemical shifts of A2AAR in the two different environments.

Figure 3 is of minimal interest, it only shows that the sample still has a signal after being stored at 4°C.

What is the variation of signal intensity between the two spectra? This would help us understand how much protein is being lost.

Response:

The long-term stability of GPCR preparations is of great interest for experimental studies such as multi-dimensional NMR experiments. We follow the suggestion by the reviewer to add a quantitative comparison of the signal intensities between the two spectra 2b and 3a, and added the new Figure 3b, which shows (jointly with Figure 3a) at the detail afforded by 2D NMR that A2AAR is indeed preserved after 1 month.

Panel a of Figure 4 should be omitted. It is not necessary to show the histogram.

Response:

Statistical data from the Biological Magnetic Resonance Bank provides the experimentally observed proton chemical shift distribution of tryptophan indole 15N–1H signals. Despite the unusual upfield position of the proton chemical shift of the tryptophan 2466.48 indole 15N–1H signal, it falls within the statistical range. In the interest of non-specialist readers, we prefer to keep the Figure 4a.

In 4b, in the traces, there are problems in the baseline (or phase correction), which in turn would cause problems when calculating the intensity at half height.

Response:

We appreciate that the reviewer had a close look at this figure. We re-phased the spectra and replaced the panels in Figure 4b, with the resulting improved cross sections.  

Reviewer 2 Report

The manuscript by Guo et al. describes the NMR characterization of an integral membrane protein, the human A2A adenosine receptor 18 (A2AAR) incorporated into lipid nanodiscs, with an inverse agonist molecule. The obtainment of high-quality NMR data of integral membrane proteins in detergent-free environments is an important advancement towards their study in native-like conditions. GPCRs have crucial physiological functions and are important drug targets.

The study appears sound and well-conducted. The described data is of high quality and the presentation is very clear.  

We recommend publication of the manuscript essentially as is, with just two minor modifications:

Figure 2. What are the dotted rectangular boxes? Please specify in the caption.

Figure 3. The name of the protein sample displayed in the figure now includes the isotope-labeling scheme, contrary to what is shown in figure 2b, however they are the same sample. Please uniform the names.

Author Response

We thank the three reviewers for the careful reading of our manuscript. We appreciate the criticisms and comments that have been advanced, and we hope that our detailed response in this letter can convince the editor of the importance of our work and result in acceptance of the paper for publication as an article in Molecules.

Reviewer 2

The manuscript by Guo et al. describes the NMR characterization of an integral membrane protein, the human A2A adenosine receptor (A2AAR) incorporated into lipid nanodiscs, with an inverse agonist molecule. The obtainment of high-quality NMR data of integral membrane proteins in detergent-free environments is an important advancement towards their study in native-like conditions. GPCRs have crucial physiological functions and are important drug targets.

The study appears sound and well-conducted. The described data is of high quality and the presentation is very clear.  

We recommend publication of the manuscript essentially as is, with just two minor modifications:

Figure 2. What are the dotted rectangular boxes? Please specify in the caption.

Response:

We added the following text to the captions of Figures 2 and 3a: “The spectral region outlined by broken lines in the spectra of Figure 2 contains the area that usually includes the signals of the Trp indole 15N–1H groups: it has been extended along the 1H chemical shift axis to higher field so as to include also the signal of Trp 246 at its unusual 1H chemical shift (see also Figure 4a).”

Figure 3. The name of the protein sample displayed in the figure now includes the isotope-labeling scheme, contrary to what is shown in figure 2b, however they are the same sample. Please uniform the names.

Response:

We thank the reviewer for pointing this out to us. We removed the lettering in the Figures 2a, 2b, 3a and 4b, since it was redundant with the figure legends.

Reviewer 3 Report

This manuscript " 2D NMR Spectroscopy of the G Protein-coupled Receptor A2AAR in Lipid Nanodiscs " provides a compelling and informative breakdown of the structural and dynamic features of human G protein-coupled receptors (GPCRs) when integrated into lipid nanodiscs and LMNG/CHS micelles. The authors provide detailed analysis of the 15N-1H TROSY correlation spectra of the A2A adenosine receptor in both environments, confirming that it retains its global fold and suggesting that tryptophan side chains may serve to stabilize the receptor in lipid bilayer membranes. The authors' methods offer vital insight into GPCR functional dynamics and structure-function relationships that can help direct continued drug development, making this a highly relevant and timely article. Overall, this work provides a comprehensive and engaging look at GPCRs in lipid bilayer environments, demonstrating the importance of studying these features for improved drug development.

I have nothing against the present work which was conducted in a very correct way. I validate the results but I question the usefulness of knowing the conformations in POPC/POPS.  But the membrane model is very simplistic. It would have been interesting to place charged lipids, sphingolipids, cholesterol to better mimic the membrane and thus find more marked effects.
Authors could discuss more details how the lipid nanodisc environment has been found to have a strong influence on the local structure and dynamics of the receptor and how this contributes to its overall function.
 It could also discuss how understanding the structure and dynamics of GPCRs in different lipid bilayers can inform future drug development.

Author Response

We thank the three reviewers for the careful reading of our manuscript. We appreciate the criticisms and comments that have been advanced, and we hope that our detailed response in this letter can convince the editor of the importance of our work and result in acceptance of the paper for publication as an article in Molecules.

Reviewer 3

This manuscript " 2D NMR Spectroscopy of the G Protein-coupled Receptor A2AAR in Lipid Nanodiscs " provides a compelling and informative breakdown of the structural and dynamic features of human G protein-coupled receptors (GPCRs) when integrated into lipid nanodiscs and LMNG/CHS micelles. The authors provide detailed analysis of the 15N-1H TROSY correlation spectra of the A2A adenosine receptor in both environments, confirming that it retains its global fold and suggesting that tryptophan side chains may serve to stabilize the receptor in lipid bilayer membranes. The authors' methods offer vital insight into GPCR functional dynamics and structure-function relationships that can help direct continued drug development, making this a highly relevant and timely article. Overall, this work provides a comprehensive and engaging look at GPCRs in lipid bilayer environments, demonstrating the importance of studying these features for improved drug development.

I have nothing against the present work which was conducted in a very correct way. I validate the results but I question the usefulness of knowing the conformations in POPC/POPS.  But the membrane model is very simplistic. It would have been interesting to place charged lipids, sphingolipids, cholesterol to better mimic the membrane and thus find more marked effects.

Authors could discuss more details how the lipid nanodisc environment has been found to have a strong influence on the local structure and dynamics of the receptor and how this contributes to its overall function.
 It could also discuss how understanding the structure and dynamics of GPCRs in different lipid bilayers can inform future drug development.

Response:

Overall, we want to refer to our response to reviewer 1 (see next paragraph). We gladly notice that the reviewer 3 immediately sees exciting applications of the presently described technical achievement to compare A2AAR conformations in different membrane mimetics at the high detail afforded by 2D NMR, and to conduct investigations at the forefront of membrane protein structural biology.

We fully understand that the reviewer identifies limitations in the scope of our publication, and all the more we are glad to read that he sees the interest of our current results with regard to continued research on structure–function studies GCPRs and possible other membrane proteins. In this regard, we have probably not given sufficient emphasis to the importance of the observation (Figure 2, Table 1) that LMNG/CHS micelles provide an environment for A2AAR that is surprisingly close to the environment in nanodiscs. This has not been shown previously on the level of detail that is afforded by multi-dimensional NMR experiments. In further response to the reviewers’ comment, we have added a sentence to the abstract of the paper (Lines 22–25 in the manuscript).

Round 2

Reviewer 1 Report

The document has not improved since the first version. It is still just a technical report, showing that the system can be used for future experiments. It still lacks scientific interest; it does not provide any new insight into the system.

Author Response

Please see the response to the academic editor below.

Academic Editor Notes

Dear authors,
Thanks for your revised version of the paper which essentially satisfies all the reviewers' concerns.
However, before final acceptance, authors are requested to include some statements about which could be the possible applications and advance into the field given by the new observations. In fact, although a "technical" improvement could deserve to be published, authors should clearly state which is the scientific significance of studying this system; why are they studying this molecular system?
Simply affirming "we reconstituted doubly–labeled A2AAR into phospholipid nanodiscs for NMR studies" seems to be too reductive.

We appreciate your request and added text in the form of a second paragraph in 4. Conclusions. We also added the new references 51 to 55. The new text reads:

The vast majority of NMR studies on structure and function of GPCRs has so far used extrinsic probe molecules that are typically attached to amino acid side chains near the surface of the receptors [51]. The present work prepares the way to add many-parameter NMR characterization of GPCR activation in a detergent-free environment to these single-parameter approaches. The use of A2AAR as a vehicle for this technical development provides interesting perspectives for future work in cancer research. A2AAR has been shown to be the major target for influencing immunosuppressive effects of adenosine in tumor microenvironments [52, 53]. Detailed structural characterization of A2AAR is an ongoing part of projects aimed at developing new cancer drugs that target this GPCR [54, 55].